# Impact of *IKZF1* Deletions in the Prognosis of Childhood Acute Lymphoblastic Leukemia in Argentina

**DOI:** 10.3390/cancers14133283

**Published:** 2022-07-05

**Authors:** María Sara Felice, Patricia Laura Rubio, Jorge Digiorge, Mariángeles Barreda Frank, Celeste Sabrina Martínez, Myriam Ruth Guitter, Elisa Olga Sajaroff, Cristian Germán Sánchez La Rosa, Carla Luciana Pennella, Luisina Belén Peruzzo, María Alejandra Deu, Elizabeth Melania Alfaro, María Constanza Guardia, Gladys Gutierrez, María Angelica Fernández Barbieri, Ezequiel Recondo, María Soledad Vides Herrera, Vanina Livio, Constanza Arnaiz, Carolina Romero, Cristina Noemi Alonso, Jorge Gabriel Rossi

**Affiliations:** 1Hematology and Oncology Department, Hospital de Pediatría Juan P. Garrahan, Buenos Aires 1245, Argentina; patrirubio13@gmail.com (P.L.R.); jorgedigiorge@gmail.com (J.D.); ma.barredafrank@gmail.com (M.B.F.); celestesmartinez@gmail.com (C.S.M.); myriamguitter@yahoo.com (M.R.G.); cgsdoc@yahoo.com.ar (C.G.S.L.R.); carlapen@gmail.com (C.L.P.); luisinaperuzzo@gmail.com (L.B.P.); alejandradeu86@gmail.com (M.A.D.); elizabeth2006_alfaro@yahoo.com.ar (E.M.A.); 2Immunology and Rheumatology Department, Hospital de Pediatría Juan P. Garrahan, Buenos Aires 1245, Argentina; sajaroffelisa@yahoo.com (E.O.S.); jrossi@garrahan.gov.ar (J.G.R.); 3Hematology and Oncology Department, Hospital del Niño Jesús, San Miguel de Tucumán, Tucumán 4000, Argentina; guardiamane@gmail.com; 4Hematology and Oncology Department, Hospital Juan Pablo II, Corrientes 1435, Argentina; paloma2578@hotmail.com; 5Hematology and Oncology Department, Hospital de Niños de San Isidro, Buenos Aires 1245, Argentina; angiemocorrea@gmail.com; 6Hematology and Oncology Department, Hospital Nacional de Clínicas San Martín, Buenos Aires 1245, Argentina; sqrecondo@yahoo.com; 7Hematology and Oncology Department, Hospital Eva Perón, Catamarca 1435, Argentina; marisolvid@hotmail.com; 8Hematology and Oncology Department, Hospital Avelino Castelán, Resistencia, Chaco 3508, Argentina; vanilivio@hotmail.com; 9Hematology and Oncology Department, Hospital de Niños Castro Rendón, Neuquén 8300, Argentina; constanzaarnaiz@gmail.com; 10Hematology and Oncology Department, Hospital Alexander Fleming OSEP, Mendoza 5500, Argentina; carolromero81@yahoo.com.ar; 11Area of Specialized Laboratories, Hospital de Pediatría Prof. Dr. Juan P. Garrahan, Buenos Aires 1245, Argentina; crisalon@gmail.com

**Keywords:** acute lymphoblastic leukemia, children, IKZF1 deletion, prognosis

## Abstract

**Simple Summary:**

An association of deletions in the *IKZF1* gene (IKZF1del) with poor prognosis in acute lymphoblastic leukemia (ALL) has been demonstrated. However, the co-occurrence of IKZF1del with deletions in other genes (IKZF1plus) seems to better define prognosis. Our aim was to analyze the influence of IKZF1del and/or IKZF1plus profiles on the survival of children with ALL. We analyzed 1023 eligible cases who were classified into three subsets: IKZF1not-del: IKZF1-not-deleted (*n* = 835), IKZF1del: IKZF1deleted (*n* = 94) and IKZF1plus: IKZF1del plus deletion of other genes (*n* = 94). Leukemia-free-survival probability (standard deviation) (LFSp(SE)) was 75 (2)% for IKZF1not-del, 51 (6)% for IKZF1del and 48 (6)% for IKZF1plus (*p*-value < 0.00001). The LFSp(SE) according to ALL-IC risk-group stratification showed a statistically significant difference within Intermediate-Risk patients, LFSp (SE) being 77 (2)% for IKZF1not-del, 61 (10)% for IKZF1del and 54 (7)% for IKZF1plus (*p*-value = 0.0005). The LFSp(SE) of the high-risk group was: 61 (4)% IKZF1not-del, 38 (8)% IKZF1del and 35 (9)% IKZF1plus (*p*-value = 0.0102). Thus, the *IKZF1* status was shown to define a population of patients with a poor outcome, mainly in those with intermediate risk. However, not-significant differences were observed between the LFSp of IKZF1del versus IKZF1plus groups. The study of the *IKZF1* status should be incorporated into the risk-group stratification of pediatric ALL.

**Abstract:**

An association of deletions in the *IKZF1* gene (IKZF1del) with poor prognosis in acute lymphoblastic leukemia (ALL) has been demonstrated. Additional deletions in other genes (IKZF1plus) define different IKZF1del subsets. We analyzed the influence of IKZF1del and/or IKZF1plus in the survival of children with ALL. From October 2009 to July 2021, 1055 bone marrow samples from patients with ALL were processed by Multiplex ligation-dependent probe amplification (MLPA). Of them, 28 patients died during induction and 4 were lost-in-follow-up, resulting in an eligible 1023 cases. All patients were treated according to ALLIC-BFM-2009-protocol. Patients were classified into three subsets: IKZF1not-deleted (IKZFF1not-del), IKZF1deleted (IKZF1del) and IKZF1del plus deletion of *PAX5, CDKN2A, CDKN2B* and/or alterations in *CRLF2* with *ERG*-not-deleted (IKZF1plus). The LFSp and SE were calculated with the Kaplan–Meier calculation and compared with a log-rank test. From the 1023 eligible patients, 835 (81.6%) were defined as IKZF1not-del, 94 (9.2%) as IKZF1del and 94 (9.2%) as IKZF1plus. Of them, 100 (9.8%) corresponded to Standard-Risk (SRG), 629 (61.5%) to Intermediate-Risk (IRG) and 294 (28.7%) to High-Risk (HRG) groups. LFSp(SE) was 7 5(2)% for IKZF1not-del, 51 (6)% for IKZF1del and 48 (6)% for IKZF1plus (*p*-value < 0.00001). LFSp(SE) according to the risk groups was: in SRG, 91 (4)% for IKZF1not-del, 50 (35)% IKZF1del and 100% IKZF1plus (*p*-value = ns); in IRG, 77 (2)% IKZF1not-del, 61 (10)% IKZF1del and 54 (7)% IKZF1plus (*p*-value = 0.0005) and in HRG, 61 (4)% IKZF1not-del, 38 (8)% IKZF1del and 35 (9)% IKZF1plus (*p*-value = 0.0102). The *IKZF1* status defines a population of patients with a poor outcome, mainly in IRG. No differences were observed between IKZF1del versus IKZF1plus. MLPA studies should be incorporated into the risk-group stratification of pediatric ALL.

## 1. Introduction

Almost 80% of pediatric patients with B-cell precursor (Bcp) acute lymphoblastic leukemia (ALL) treated with modern protocols can achieve a long-term cure in Argentina [1]. However, a significant proportion of these patients still experience relapse and therapy-related toxicities. Thus, the focus of therapy improvement for children with ALL, or any childhood cancer, is not only to cure a higher number of patients, but also to minimize both short-term and long-term therapy-associated toxicities. In this line of thought, we aim for a better stratification of patients in order to achieve a more accurate adaptation of the treatment.

The presence of acquired chromosomal and genetic abnormalities in the leukemic cells of patients with ALL is one of the principal hallmarks of the disease. Over the past decade, numerous structural and numerical aberrations have been discovered and characterized in ALL. These anomalies are leukemia-specific and are used to diagnose and classify the disease [2,3,4,5,6]. One of these genetic abnormalities is the alteration in the number of copies of different genes, and *IKZF1* is one of the genes that has been described with prognostic impact [7,8]. 

The *IKZF1* gene encodes Ikaros, a zinc-finger transcription factor required for the development of all lymphoid lineages. Somatic deletions of *IKZF1* have been described as a new high-risk marker in Bcp ALL [9,10]. In recent years, several authors demonstrated that the activation of JAK-STAT signaling may enhance, whereas deletions of *ERG* gene can attenuate, the negative prognostic effect conferred by *IKZF1* deletions in Bcp ALL [11,12]. The association of deletions in the *IKZF1* gene (IKZF1del) with poor prognosis in ALL has been demonstrated. In addition, deletions in other genes such as *PAX5, CDKN2A, CDKN2B* and alterations in *CRLF2*, in absence of the deletion of the *ERG* gene have been defined by Stanulla et al. as a new *IKZF1* subtype (named IKZF1plus) with a worse prognostic impact than IKZF1del cases [13].

The different genetic subtypes of ALL should be correlated with the response to treatment of these patients, and these two factors generally define risk-group stratification. Early prednisone response and minimal residual disease (MRD) are two of the most important parameters considered for assessing treatment response in BFM-based protocols.

Our aim was to retrospectively analyze the influence of IKZF1del and/or IKZF1plus in the survival rates of children who were administered the ALLIC protocol in several centers in Argentina, members of the Argentinean Society of Pediatric Hematology and Oncology (SAHOP).

## 2. Materials and Methods

This is an observational, retrospective analysis study. From October 2009 to July 2021, 1055 bone marrow samples from consecutive and homogeneously treated patients with ALL were processed by multiplex ligation-dependent probe amplification (MLPA) and retrospectively analyzed. Of them, 28 patients died during induction and 4 were lost in follow-up, thus resulting in a final number of 1023 cases eligible for this study. 

Diagnosis of leukemia was based on morphological findings with the confirmation of the presence of lymphoblasts by flow cytometry according to the BFM study group guidelines. All cases were also evaluated by G-banding and RT-PCR following the recommendations previously described for detection of recurrent fusion genes: *ETV6–RUNX1*, *BCR–ABL1* (p190 and p210), *KTM2A–AFF1* and *TCF3–PBX1* [14].

All patients were stratified and treated according to ALLIC-BFM-2009-protocol criteria. This protocol classifies ALL patients in 3 risk groups according to age at diagnosis, initial WBC count, genetic recurrent alterations, response to prednisone on day 8 of treatment and MRD by flow cytometry on day 15 of treatment. 

Response was evaluated at different time-points according to the protocol criteria, and complete remission (CR) was defined with conventional criteria of less than 5% of blast in bone marrow, with hematopoietic regeneration of all lineages and no evidence of leukemia compromise in extra-medullar sites. CR was assessed at the end of induction (day 33) and at the end of consolidation (day 78) in patients who did not achieve CR on day 33. Null response was defined as patients who did not achieve CR at the end of consolidation.

Patients were stratified as Standard (SRG), Intermediate (IRG) and High Risk (HRG), based on the aforementioned parameters. The risk-group distribution of the evaluable patients was: 100 SRG, 629 IRG and 294 HRG. The biological features and response to treatment of the different subgroups of 1023 eligible patients were compared.

MRD on day 15 was defined as positive with a cut-off ≥ 0.1%. MRD was also analyzed on day 33 (end of induction) and day 78 (end of consolidation) of treatment by flow cytometry in most of the patients, although no clinical decisions were taken based on these data. Positive MRD was defined as ≥ 0.05% at these time-points for correlating MRD and the outcome of *IKZF1* subsets of patients.

MLPA assay was performed for all samples using two kits: Salsa MLPA Probemix P335 ALL-IKZF1 (MRC-Holland, Amsterdam, The Netherlands) to characterize copy number alterations of *IKZF1, PAX5, ETV6, RB1, BTG1, EBF1, CDKN2A-CDKN2B* genes and PAR1 region (*SHOX* area, *CRLF2, CSF2RA, IL3RA* and *P2RY8* genes); and Salsa MLPA Probemix P327 iAMP21-ERG (MRC-Holland, Amsterdam, the Netherlands) to identify copy number alterations of *RUNX1* and *ERG* genes. Additionally, *P2RY8-CRLF2* alteration was confirmed by RT-PCR [10,12]. 

Based on MLPA analyses and for the purpose of the present study, patients were classified into 3 subsets: IKZF1not-deleted (IKZF1not-del), IKZF1deleted (IKZF1del) and IKZF1del plus deletion of *PAX5, CDKN2A, CDKN2B* and/or *CRLF2* with *ERG*-not-deleted (IKZF1plus). It is important to mention that IKZF1 status was not included in the criteria for stratifying patients into ALL-IC risk groups.

We also analyzed the impact of these 3 genetic subgroups within the different ALL-IC risk groups.

Comparison of the biological characteristics for discrete variables was assessed with χ^2^ test and for continuous variables Wilcoxon Rank Sum Test was used. Leukemia Free Survival Probability (LFSp) and Standard Error (SE) were analyzed using Statistix 7.1 software, evaluated with the Kaplan–Meier calculation, and compared with the log-rank test [15,16].

## 3. Results

### 3.1. Analysis of Biological Features and Classification of Patients 

From the 1023 eligible patients for this study, 835 (81.6%) were defined as IKZF1not-del, 94 (9.2%) as IKZF1del and 94 (9.2%) as IKZF1plus according to *IKZF1* status.

Initially, we correlated the findings of *IKZF1* subgroups with other prognostic features: age, initial WBC count, immunophenotype and other recurrent abnormalities. These findings are shown in Table 1.

When biological features were compared, no significant differences were observed in WBC count at the moment of diagnosis between IKZF1not-del and IKZF1del, but IKZF1plus disclosed a significantly higher WBC count than both IKZF1not-del and IKZF1del. The comparison of the presence of genetic abnormalities showed a significantly higher incidence of *BCR–ABL1* fusion transcript and hypodiploidy in IKZF1not-del vs. IKZF1del, but the difference remained significant only for *BCR–ABL1* when the comparison was made between IKZF1plus vs. IKZF1not-del plus IKZF1del. The incidence of *ETV6–RUNX1* was significantly superior in IKZF1not-del vs. IKZF1del, and remained higher when the comparison was made between IKZF1not-del plus IKZF1del vs. IKZF1plus.

The HRG rate was higher in IKZF1del vs. IKZF1not-del and when comparing IKZF1plus vs. IKZF1not-del plus IKZF1del.

### 3.2. Evaluation of Treatment Response and Survival

We also analyzed the response to treatment and MRD results in this population of patients considering *IKZF1* subsets. MRD on day 15 was available in 952 cases, on day 33 in 905 cases and on day 78 in 926 patients. Of them, on day 15, 665 cases were positive and 287 were negative. A significantly higher incidence of cases with positive day 15-MRD was observed in IKZF1del when compared with IKZF1not-del, but no differences were observed between IKZF1plus vs. IKZF1not-del and IKZF1del.

MRD analysis on day 33 showed 738/905 positive and 167/905 negative results and MRD on day 78 was negative in 873/926 cases and positive in 53/926 cases. The number of positive MRD cases was significantly higher at both time-points of evaluation for IKZF1del vs. IKZF1not-del and when comparing IKZF1plus vs. IKZF1not-del and IKZF1del. 

The LFSp (SE) was 75 (2)% for IKZF1not-del, 51 (6)% for IKZF1del and 48 (6)% for IKZF1plus (*p*-value < 0.00001) (Figure 1A).

The LFSp (SE) according to risk groups was: in SRG, 91 (4)% for IKZF1not-del, 50 (35)% for IKZF1del and 100% for IKZF1plus (*p*-value = ns) (Figure 1B); in IRG, 77 (2)% for IKZF1not-del, 61 (10)% for IKZF1del and 54 (7)% for IKZF1plus (*p*-value = 0.0005) (Figure 1C); and in HRG, 61 (4)% for IKZF1not-del, 38 (8)% for IKZF1del and 35 (9)% for IKZF1plus (*p*-value = 0.0102) (Figure 1D).

When we compared the outcome of the different *IKZF1* subsets according to MRD results on day 33, a statistically significant difference was observed between the three groups in the population of patients with negative MRD, the LFSp (SE) being 79 (2)% for IKZF1not-del, 62 (9)% for IKZF1del and 56 (7)% for IKZF1plus (*p*-value = 0.0003) (Figure 2A). In the group of patients with positive MRD at that time-point, LFSp (SE) was 50 (6)% for IKZF1not-del, 41 (9)% for IKZF1del and 38 (3)% for IKZF1plus, respectively (*p*-value = 0.3192) (Figure 2B).

The analysis of the impact of *IKZF1* subgroups according to MRD on day 78 disclosed, for the population with negative MRD, an LFSp (SE) of 77 (2)% for IKZF1not-del, 60 (7)% for IKZF1plus and 64 (6)% for IKZF1del (*p*-value = 0.0135) (Figure 2C). The group of patients with positive MRD on day 78 showed an LFSp (SE) of 37 (9)% for IKZF1not-del, 25 (14)% for IKZF1del and 0% for IKZF1plus (*p*-value = 0.3225) (Figure 2D).

## 4. Discussion

The prognosis of childhood ALL has improved with the current treatment protocols, achieving 90% survival rate in the different referral international groups. However, in our country, EFS rates reach almost 80%, and all efforts should be directed to identify the means for curing a higher number of children with this disease [1,17,18]. The difference in EFS rates in Argentina is probably related to several factors. One of them could be the need for a better characterization of the biological features of the leukemia cells at the moment of initial diagnosis. The other factor to consider, and to include in future protocols, is the evaluation of MRD in additional time-points, such as day 33 and day 78, which are not used for stratification in the current protocol. Finally, Argentina is a large country and children with ALL are treated in approximately 30 different treatment centers. The support measures are heterogeneous in the different centers and regions and need to be improved in order to decrease the death rates during the induction phase and in CR, so as to achieve equal standards and possibilities of curing the disease in every center of the country. All these points are the challenges facing upcoming studies.

The two most powerful factors associated with prognosis of pediatric ALL are the genetic abnormalities and the response to treatment, assessed through MRD measurements and thus, taking into account both factors seems to be the best way to achieve a more accurate treatment tailoring.

The identification of new genetic–molecular abnormalities with prognostic impact is therefore the subject of several investigations. The deletion of *IKZF1* gene has been associated with poor prognosis in children and adults with ALL diagnosis [7,8,9,10]. Over the last few years, Stanulla et al. have reported a new subset of cases with poorer prognosis within the group of ALL patients with the deletion in the *IKZF1* gene. This subgroup has been named “IKZF1plus” [13]. The reported correlation of the different *IKZF1* subsets (*IKZF1* not-del, IKZF1del and IKZF1plus) within the risk groups, according to BFM criteria, showed that both the IKZF1not-del group and IKZF1del group achieved similar EFS probability, while IKZF1plus achieved a significantly inferior EFS probability due to an increased incidence of relapses. These results were confirmed in the same analysis performed by the AIEOP group and presented as well in the same publication [13].

We analyzed our population of 1023 eligible cases, homogeneously treated over almost 12 years, detecting a similar incidence of *IKZF1* subsets through MLPA evaluation. The subgroups of patients with IKZF1del and IKZF1plus disclosed a significantly higher incidence of biological features associated with poor outcome than those without IKZF1 deletion: a higher median of WBC count at diagnosis, a higher incidence of recurrent genetic abnormalities such as *BCR–ABL1* fusion transcript and/or a lower incidence of *ETV6–RUNX1* transcript. All these associations explain the higher rate of HRG patients in the population of IKZF1del and IKZF1plus subgroups, as previously reported [19].

The number of patients with positive MRD on day 33 and on day 78 was also higher in the subsets IKZF1del and IKZF1plus, as well as the number of patients who did not achieve CR at the end of induction.

Based on this distribution of prognostic features, we observed a statistically significant difference in the LFS rates according to these *IKZF1* subsets. The main adverse event was relapse of ALL. However, when we compared the relapse rates according to *IKZF1* subgroups, they were significantly higher in the subgroups with *IKZF1* deletion than they were in patients without *IKZF1* deletion, and the difference was greater when we compared patients without IKZF1 deletion versus both populations with *IKZF1* deletions (IKZF1del and IKZF1plus).

Regarding the analysis of risk groups, according to protocol stratification, our results agree with previous reports [13] with respect to the influence of *IKZF1* deletion in the survival rates of patients in IRG and HRG, but not in the SRG, supporting the relevance of a good, early response, even in the group of patients with adverse genetic abnormalities.

When we correlated the impact of *IKZF1* deletion with MRD on day 33 and on day 78, which had not been considered for the risk-group stratification of these patients, we observed a clear influence of *IKZF1* deletion in the patients with negative MRD at both time-points, and when MRD was positive, no significant differences were observed in the survival rate. This fact reaffirms the adverse prognosis conferred by a slower and later clearance of the blasts, measured through MRD, for all ALL genetic–molecular subtypes, as previously described by other authors [20].

It is important to keep in mind that the first reports about the new genetic subsets of ALL [6,7,8] describe the adverse prognostic impact of *IKZF1* deletion independently of the association with additional deletions in other genes. Nowadays, some groups like the Dutch group and the ALLtogether group define HRG patients as children with ALL diagnosis showing *IKZF1* deletion only. This means that the role of *IKZF1* deletion needs additional analysis and further prospective studies in order to achieve a final definition of its prognostic impact on childhood ALL.

Therefore, this fact probably means that we still need to identify other, still unrecognized abnormalities which could anticipate the occurrence of late responders in order to refine the stratification of patients, so as to achieve a better treatment adequation. Nevertheless, chemotherapy probably has a limit for improving these results and the role of new target therapies directed to these abnormalities and pathways involved in the leukemogenesis process should be considered.

## 5. Conclusions

We conclude that *IKZF1* status defines populations of patients with significantly different survival rates, mainly in IRG. In our setting, no differences were observed between IKZF1del versus IKZF1plus subsets, mainly due to late relapses in the group of IKZF1del (not plus). The correlation of the IKZF1 subset with MRD on day 33 and on day 78 helps to attain a better definition of patient subsets. Copy number alteration studies such as MLPA should be incorporated as a parameter for risk-group stratification of pediatric ALL, in order to achieve a better adjustment of treatment intensity.

## Figures and Tables

**Figure 1 cancers-14-03283-f001:**
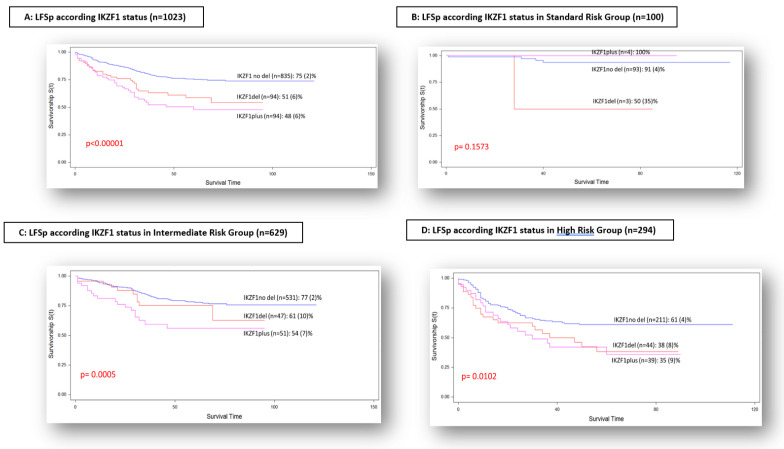
LFS probability according to *IKZF1* status and according to risk-group stratification.

**Figure 2 cancers-14-03283-f002:**
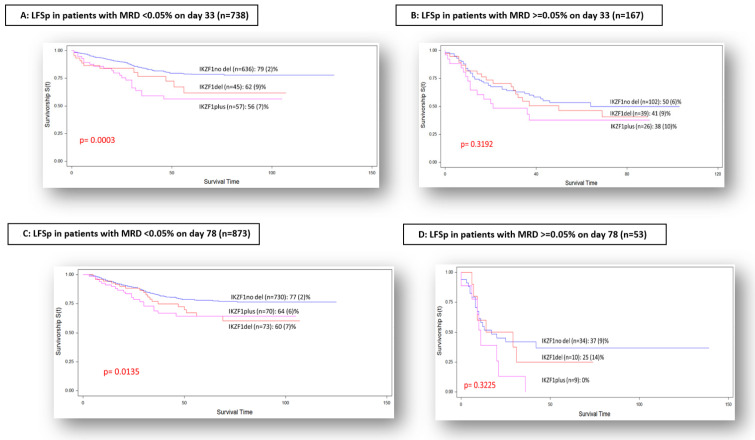
LFS probability according to MRD on day 33 and on day 78.

**Table 1 cancers-14-03283-t001:** Comparison of biological features and response to treatment according to *IKZF1* status.

Characteristics	No *IKZF1* Deleted (*n* = 835)	*IKZF1* Deleted (*n* = 94)	*IKZF1* Plus (*n* = 94)	*p*-Value (*IKZF1* Del vs. Not Del)	*p*-Value (*IKZF1* Del + Not Del vs. *IKZF* Plus)
Mean age (in years) (range)	6 (2 mo–16 y)	8 (1y–16y)	8 (11 mo–16 y)	Ns	Ns
Mean WBC (range)/mm^3^	57,196 (1170–668,000)	59,164 (560–668,000)	99,581 (1680–720,000)	Ns	<0.00001
Immunophenotype B/T/AMBI	718/108/9	89/4/1	88/2/4	Ns	Ns
*BCR–ABL1*	13	11	10	<0.00001	<0.00001
Hypodiploidy	7	3	1	0.0361	Ns
*KTM2A*r	48	3	3	Ns	Ns
*ETV6–RUNX1*	134	3	5	0.0009	0.0118
PPR	116 (13.9%)	17 (18.0%)	15 (16.1%)	Ns	Ns
MRD on day 15 (>0.1%)	517/772 (66.9%)	86/92 (93.5%)	62/84 (73.8%)	<0.00001	Ns
MRD on day 33+ (≥0.05%)	102/738 (13.8%)	39/84 (46.4%)	26/83 (31.3%)	<0.00001	0.0018
MRD on day 78+ (≥0.05%)	34/764 (4.5%)	10/83 (12.0%)	9/79 (11.4%)	0.0045	0.0437
ALLIC risk group distribution					
Standard Risk	93 (11%)	3 (3%)	4 (4%)		
Intermediate Risk	531 (64%)	47 (50%)	51 (54%)		
High Risk	211 (25%)	44 (47%)	39 (42%)		
Null response (day 33)	6 (0.7%)	9 (9.6%)	7 (7.4%)	<0.00001	0.0002
Relapses	117 (14.0%) (2 SMN)	24 (25.5%) (1 SMN)	27 (28.7%)	0.0032	0.0007
Mean time (range) to relapse (in months)	24 (2–75)	27 (4–68)	23 (2–59)	Ns	Ns

Abbreviations: AMBI: ambiguous lineage leukemia; *KTM2A*r: KTM2A rearrangements; PPR: prednisone poor responders; MRD: minimal residual disease; SMN: second malignant neoplasm; Ns: not significant; y: years-old; mo: months-old.

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
