# Peer review of "Impact of IKZF1 Deletions in the Prognosis of Childhood Acute Lymphoblastic Leukemia in Argentina"

_cancers, 2022, doi:10.3390/cancers14133283_

Round 1

Reviewer 1 Report

Maria S. Felice et al. Impact of IKZF1 deletion ...

In this paper the authors investigate the impact of the so-called IKFZplus group previously established by Stanulla et al. for the outcome of children with cALL in Argentina in comparison with other patients harboring or not IKZF1 deletions taking in consideration also the risk stratification based on response to prednisone at day 8 and MRD at day 15, 33, and 78. For that the authors studied retrospectively a cohort of more than 1000 patients with about 90 harboring deletion in IKZF1 (IKZF1del or group 2) and about 90 harboring deletion IKZF1 and other well defined factors (IKZF1plus or group 3). All other patients were the group-1.

The paper is a very important piece of work and for the cALL patient cohort of childhood cALL in Argentina and South America in general very important. Unfortunately, the manuscript is flawed by a very confusing text, which needs a better structure and organization. Also more figures and better explanation of the results are needed. To this reviewer it seems that there are much more results in these data than shown in this paper. In addition, there is the urgent need to address the spelling and grammar errors. An in depth review by a native speaker is recommended.

Major points:

1.) simple Abstract. The simple abstract is not simple at all. It has to be understood by people not familiar with clinical trials and childhood ALL. Also abbreviations have too be avoided or explained.

2.) Abstract. abbreviations have to be explained. Not everybody knows what MLPA means.

3.) Introduction. "..our country" - Argentina? Otherwise the Introduction gives a good overview over the topic of the paper.

4.) Mat/Meth. A better outline of the overall patient characteristics is needed. Maybe introduced in the result section? Does this correspond to the normal distribution in other studies? Less explanation of the groups (it is repeated in the result section), more about the patients cohorts (biases, retrospective, etc.). Maybe a work flow about the sequence of methodologies in a figure could help. A better explanation of the techniques would make things clearer.

5.) Results. The overall description of the results is confusing. Is the presence of IKZF1 deletion of the cohort in accordance with that published by other groups etc?

  • less repetition (overall the text!). The definition of the groups is explained too often.

  • a clearer structure of the analyses is needed. If the three groups are to be investigated keep them separated. First WBC and the three groups, then the genetic aberration in the three groups, MRD and the three groups, risk stratification and.....

  • what happened with a MRD-positive at day 8 or 15?

6.) Discussion. This reviewer would introduce other points: e.g. why the differences between Argentina and other countries regarding EFS. Less repetition. No need to cite again Stanulla et al and the definition of the groups based on IKZF1 deletions. A bit less recapitulation of the results and more consideration for them in the context of the literature.

Author Response

Reviewer 1:

Comments and Suggestions for Authors

Open Review

English language and style

( ) Extensive editing of English language and style required
(x) Moderate English changes required
( ) English language and style are fine/minor spell check required
( ) I don't feel qualified to judge about the English language and style

Yes

Can be improved

Must be improved

Notapplicable

Does the introduction provide sufficient background and include all relevant references?

(x)

( )

( )

( )

Is the research design appropriate?

( )

(x)

( )

( )

Are the methods adequately described?

( )

( )

(x)

( )

Are the results clearly presented?

( )

( )

(x)

( )

Are the conclusions supported by the results?

(x)

( )

( )

( )

Maria S. Felice et al. Impact of IKZF1 deletion ...

In this paper the authors investigate the impact of the so-called IKFZplus group previously established by Stanulla et al. for the outcome of children with cALL in Argentina in comparison with other patients harboring or not IKZF1 deletions taking in consideration also the risk stratification based on response to prednisone at day 8 and MRD at day 15, 33, and 78. For that the authors studied retrospectively a cohort of more than 1000 patients with about 90 harboring deletion in IKZF1 (IKZF1del or group 2) and about 90 harboring deletion IKZF1 and other well defined factors (IKZF1plus or group 3). All other patients were the group-1.

The paper is a very important piece of work and for the cALL patient cohort of childhood cALL in Argentina and South America in general very important. Unfortunately, the manuscript is flawed by a very confusing text, which needs a better structure and organization. Also more figures and better explanation of the results are needed. To this reviewer it seems that there are much more results in these data than shown in this paper. In addition, there is the urgent need to address the spelling and grammar errors. An in depth review by a native speaker is recommended.

Major points:

1.) simple Abstract. The simple abstract is not simple at all. It has to be understood by people not familiar with clinical trials and childhood ALL. Also abbreviations have too be avoided or explained.

Comments and Action: the reviewer´s suggestions were taken, and the simple Abstract was reviewed and modified. 

2.) Abstract. abbreviations have to be explained. Not everybody knows what MLPA means.

Comment and Action: the abbreviation is replaced by its explanation

3.) Introduction. "..our country" - Argentina? Otherwise, the Introduction gives a good overview over the topic of the paper.

Comments and Action: the reviewer´s suggestion was taken, and the text was modified. 

4.) Mat/Meth. A better outline of the overall patient characteristics is needed. Maybe introduced in the result section? Does this correspond to the normal distribution in other studies? Less explanation of the groups (it is repeated in the result section), more about the patients cohorts (biases, retrospective, etc.). Maybe a work flow about the sequence of methodologies in a figure could help. A better explanation of the techniques would make things clearer.

Comments and Actions: Thanks for the reviewer´s suggestions. In the Methods sections it was included some population definition: for example, in the first paragraph (line 142), it was included the definition of the population of patients that was consecutively and homogeneously treated, in order to clarify that there were not biases.

The study was defined in the first line as an observational and retrospective study, too.

About the work flow, the second paragraph of the section (lines 147 to 151), defines the diagnostic studies performed to all the samples, and I do not think that a work flow would be necessary.  

On the other hand, a more detailed description of the characteristics of the population will be included in the publication of the whole ALLIC 2009 protocol, which will be presented soon.

I think the described biological and response characteristics are enough for analyze this group of patients. Please, let us know what other data the reviewer would like to know, and will be a pleasure to include them.

5.) Results. The overall description of the results is confusing. Is the presence of IKZF1 deletion of the cohort in accordance with that published by other groups etc?

  • less repetition (overall the text!). The definition of the groups is explained too often.
  • a clearer structure of the analyses is needed. If the three groups are to be investigated keep them separated. First WBC and the three groups, then the genetic aberration in the three groups, MRD and the three groups, risk stratification and.....
  • what happened with a MRD-positive at day 8 or 15?

Comments and actions: We have taken the suggestions and the text was modified.

The table 1 was improved, with the requested details (WBC, Risk groups, genetic aberrations, etc).

With regard to MRD on day 8, this protocol did not include this determination and data about MRD on day 15 is included in the table.

6.) Discussion. This reviewer would introduce other points: e.g. why the differences between Argentina and other countries regarding EFS. Less repetition. No need to cite again Stanulla et al and the definition of the groups based on IKZF1 deletions. A bit less results and more consideration for them in the context of the literature recapitulation of the

Comments and Actions: The discussion was modified according to the reviewer´s suggestions and we hope it was improved.

The first question about the differences in EFS in Argentina was answered in the first paragraph of the discussion section (line 277 to 286).

A new paragraph about the consideration of the results of our manuscript are included, as requested (line 333 to 339).

Reviewer 2 Report

In Table.1, 6 null response subjects in IKZF-1-non-del group, I suspect these subjects have point mutations in IKZF-1 ORF which cause significant functional down-regulation. It is strongly suggested that authors conduct the whole ORF sequencing analysis of IKZF1 on these 6 subjects.

Author Response

Reviewer 2

Open Review

English language and style

( ) Extensive editing of English language and style required
( ) Moderate English changes required
( ) English language and style are fine/minor spell check required
(x) I don't feel qualified to judge about the English language and style

Yes

Can be improved

Must be improved

Notapplicable

Does the introduction provide sufficient background and include all relevant references?

(x)

( )

( )

( )

Is the research design appropriate?

( )

(x)

( )

( )

Are the methods adequately described?

(x)

( )

( )

( )

Are the results clearly presented?

( )

(x)

( )

( )

Are the conclusions supported by the results?

( )

(x)

( )

( )

Comments and Suggestions for Authors

In Table.1, 6 null response subjects in IKZF-1-non-del group, I suspect these subjects have point mutations in IKZF-1 ORF which cause significant functional down-regulation. It is strongly suggested that authors conduct the whole ORF sequencing analysis of IKZF1 on these 6 subjects.

Comment: We think that this presumption is a good option for understanding the null response of these patients, and we could perform this analysis of ORF in a future. However, we can speculate that other genetic abnormalities (known and unknown) could play a role in this primary resistance to chemotherapy. We do not know all genetic abnormalities of ALL, yet.

Action: We will take the reviewer´s suggestion and we will extend the analysis of these 6 resistant cases, in the future, and it could be motive of other publication.

Final comment: English language and style was reviewed, and we hope that it has been improved.

Round 2

Reviewer 1 Report

The only thing I would add in the discussion is a comparison between Argentina and the published numbers of EFS in other publications/countries like US or Europe.One could write: "There is a discrepancy between the rural,,, and the metropolitan... centers in Argentina (as you already di - this reviewer is rather sure, that such differences also exist in the US). Overall there are.... between Argentina and the reported EFS for other countries like......(Ref)"

Author Response

The only thing I would add in the discussion is a comparison between Argentina and the published numbers of EFS in other publications/countries like US or Europe.One could write: "There is a discrepancy between the rural,,, and the metropolitan... centers in Argentina (as you already di - this reviewer is rather sure, that such differences also exist in the US). Overall there are.... between Argentina and the reported EFS for other countries like......(Ref)"

Comments: Thanks again for this comment and suggestion.

I think that the explanation to this question/comment was answered with the addition of a paragraphs in the discussion (line 222 to 231). We described in this paragraph the possible causes of the differences in the EFS between Argentina and other countries and we also defined these topics as a challenge to overcame.

In addition, I think that we should say that this differences in EFS are observed not only in Argentina, but also in Latin-American countries in general. The first refference of our manuscript (J Clin Oncol 2013.53.2754) described an EFS of 74% for the ALLIC 2002 protocol, and the ALLIC 2009 (unpublished data which will be publish soon) show a minor improvement in these results.

Regarding Argentina, there is no doughts about the differences among different regions in our country. In fact, we analyzed and published this asimetry 8 years ago (and I am attaching this paper, with the abstract in English as support of my considerations. We specially analyzed mortality of patients with malignant diseases. Acute Leukemias was the most frequent diagnosis of dead patients and most of them died after relapse. One of the causes of these differences are probably related to delays in the administration of treatment – which obviously impact in the cummulative incidence of relapses – and is also associated to differences in the support measures in the different centers and regions in the country. For this reason we are working as SAHOP (Argentinean Society of Pediatric Hematology and Oncology) and for this reason this manuscript includes al participant centers in the study. However, I think that this detailed analysis, which is matter of permanent concern, excedes this manuscript, and, as I previously said, is not a problem only in Argentina, because other countries in Latin America achieve this lower EFS when a comparison is made with reference groups or center.

Action: We change the referred lines in the answer to the Reviewer 1 and hignlighted these changes.

We do not think that the analysis about different regions of provinces in Argentina should be included in this paper, because excedes the aim or our manuscript. The mentioned paragraph (line 222 to 231) explaines the possible causes of the poorer results in Argentina. We hope our comments would be useful for clarifing this topic.

Reviewer 2 Report

They have nicely revised their manuscript with sufficient discussion. I agree with publication.